# Diabetic Ketoacidosis in Children and Adolescents; Diagnostic and Therapeutic Pitfalls

**DOI:** 10.3390/diagnostics13152602

**Published:** 2023-08-04

**Authors:** Eirini Kostopoulou, Xenophon Sinopidis, Sotirios Fouzas, Despoina Gkentzi, Theodore Dassios, Stylianos Roupakias, Gabriel Dimitriou

**Affiliations:** Department of Paediatrics, University of Patras, 26504 Patras, Greece; xsinopid@upatras.gr (X.S.); sfouzas@upatras.gr (S.F.); gkentzid@upatras.gr (D.G.); tdassios@upatras.gr (T.D.); stylroup@yahoo.gr (S.R.); gdim@upatras.gr (G.D.)

**Keywords:** diabetic ketoacidosis, type 1 diabetes mellitus, pitfalls, children, diagnosis, management

## Abstract

Diabetic ketoacidosis (DKA) represents an acute, severe complication of relative insulin deficiency and a common presentation of Type 1 Diabetes Mellitus (T1DM) primarily and, occasionally, Type 2 Diabetes Mellitus (T2DM) in children and adolescents. It is characterized by the biochemical triad of hyperglycaemia, ketonaemia and/or ketonuria, and acidaemia. Clinical symptoms include dehydration, tachypnoea, gastrointestinal symptoms, and reduced level of consciousness, precipitated by a variably long period of polyuria, polydipsia, and weight loss. The present review aims to summarize potential pitfalls in the diagnosis and management of DKA. A literature review was conducted using the Pubmed/Medline and Scopus databases including articles published from 2000 onwards. Diagnostic challenges include differentiating between T1DM and T2DM, between DKA and hyperosmolar hyperglycaemic state (HHS), and between DKA and alternative diagnoses presenting with overlapping symptoms, such as pneumonia, asthma exacerbation, urinary tract infection, gastroenteritis, acute abdomen, and central nervous system infection. The mainstays of DKA management include careful fluid resuscitation, timely intravenous insulin administration, restoration of shifting electrolyte disorders and addressing underlying precipitating factors. However, evidence suggests that optimal treatment remains a therapeutic challenge. Accurate and rapid diagnosis, prompt intervention, and meticulous monitoring are of major importance to break the vicious cycle of life-threatening events and prevent severe complications during this potentially fatal medical emergency.

## 1. Introduction

DKA represents the most common acute hyperglycaemic emergency in children and adolescents with diabetes mellitus [1]. Based on the *International Society for Pediatric and Adolescent Diabetes* (ISPAD) guidelines, it is characterized by the biochemical triad of hyperglycaemia (serum glucose > 11 mmol/L or >200 mg/dL), ketonemia (β-hydroxybutyrate concentrations > 3.0 mmol/L) and/or moderate or large ketonuria, and a high anion-gap metabolic acidaemia (venous pH < 7.3 and/or bicarbonate < 18 mmol/L) [2,3].

Clinically, DKA is characterized by dehydration, tachypnoea and Kussmaul breathing, smell of ketones in the breath, nausea, vomiting, abdominal pain, drowsiness, confusion, reduced level of consciousness and coma, which are precipitated by a variably long period of polyuria, polydipsia, and weight loss. Most children presenting with DKA are in a volume-depleted state, which, in its most severe form, results in acute tubular necrosis and potentially in acute kidney injury (AKI) [3].

DKA occurs primarily at the onset of type 1 diabetes mellitus (T1DM) as a result of absolute or relative insulin deficiency due to autoimmune destruction of the β-cells of the islet of Langerhans and concomitant elevation of counter-regulatory hormones induced by stress, such as glucagon, growth hormone, catecholamines, and cortisol. It may also occur due to uncontrolled T1DM. Lack of adequate insulin and increase in counter-regulatory hormones lead to increased glucose production by the liver and the kidney, through gluconeogenesis and glycogenolysis, and reduced peripheral glucose utilization. As a result, hyperglycaemia, hyperosmolarity, increased lipolysis, and ketogenesis occur. Hyperglycaemia and hyperketonaemia lead to osmotic diuresis, dehydration, and electrolyte loss. Acidosis is enhanced by lactic acidosis caused by hypoperfusion.

In children and adolescents, DKA commonly occurs at the initial diagnosis of T1DM, with the incidence varying from 13% to 80% in different populations [4,5,6]. It can also occur in the context of newly diagnosed T2DM, caused by impaired insulin secretion or action, or in children and adolescents with uncontrolled T2DM, also known as ketosis-prone T2DM [1].

DKA can be precipitated by any physiological stress, including infections, with urinary tract infections and gastroenteritis being the leading causes [7,8]. Poor adherence to insulin therapy and insulin pump issues, such as dislodgement or blockage of infusion sets, are also frequent causes of DKA [9]. Among children and adolescents with known T1DM, DKA mostly occurs due to insulin omission, particularly in the presence of gastrointestinal infections with vomiting [10]. Poor diabetes control, previous episodes of DKA, dysfunctional family relationships, limited access to medical care, history of psychiatric disorders, and adolescent age are also risk factors for DKA in children and adolescents [11,12].

The mortality rate of DKA in children is reported as <1% in developed countries [13], caused primarily by cerebral injuries and cerebral oedema [14]. Nonetheless, among children with T1DM, DKA is the leading cause of mortality accounting for >50% of all deaths [15].

The present review aims to raise awareness and a high index of suspicion regarding diagnostic and therapeutic pitfalls that make optimal diagnostic and therapeutic approach of this paediatric emergency rather challenging. Early identification of symptoms associated with T1DM is crucial for prompt diagnosis and prevention of DKA. Moreover, careful, timely, and accurate DKA management is important for prevention of DKA-associated complications, prolonged hospital stays, and excessive costs [16,17].

## 2. Materials and Methods

A comprehensive literature research was conducted from January 2000 onwards and articles published in peer-reviewed international journals were considered. Data were extracted from two scientific databases, Medline (Pubmed) and Scopus. Search terms included “diabetic ketoacidosis”, “diagnosis”, “management” and “children”. All articles that examined the potential difficulties and pitfalls in the diagnosis and management of DKA in children and adolescents were considered eligible for this review. The recommendations of *Preferred Reporting Items for Systematic Reviews and Meta-Analyses* (PRISMA) guideline and checklist were followed (Figure 1).

Inclusion criteria included: children < 18 years old, reference to diagnosis and/or management of DKA, human studies, papers written in English language, types of sources: reviews, editorials, original research articles. Exclusion criteria included: non-human studies, absent or inadequate methodology, papers not relevant to the topic. The articles included are summarized in Table 1.

## 3. Results

The present review summarizes the existing knowledge on the difficulties and traps in the diagnosis and management of DKA in children and adolescents. The data presented highlight the importance of raised awareness regarding the challenges in the diagnostic and therapeutic approach of DKA due to the variable clinical presentation, the overlapping symptoms with other medical diagnoses, and the complex pathophysiological mechanisms involved in the management of this potentially fatal medical condition.

## 4. Discussion

### 4.1. Pitfalls Related to the Diagnosis of DKA

DKA is more frequent in children with T1DM, but it also occurs in adolescents with T2DM [4]. Among those with T1DM, DKA is more frequent in newly diagnosed children less than 5 years old and in populations with limited access to medical care due to economic or social reasons [17,24]. In those with T2DM, a genetic predisposition for ketosis-prone T2DM is suggested by the increased incidence observed in people of African or Hispanic origin. Children and adolescents with ketosis-prone T2DM also have a strong family history of insulin resistance and T2DM, and frequently have obesity [8]. They present with decreased insulin concentrations and autoimmune markers of T1DM, such as islet cells, insulin, glutamic acid decarboxylase, and protein tyrosine phosphatase autoantibodies, at similar concentrations as those with hyperosmolar hyperglycaemic state (HHS); however, their β-cell function recovers and insulin secretion is restored soon after treatment [8]. Thus, insulin treatment is not required in the long term and oral glucose-lowering medications are appropriate. Diagnosing the type of diabetes in children and adolescents presenting with DKA can be challenging given the increased rates of obesity in the general paediatric population [25] and the positive autoimmune markers present in children and adolescents with ketosis-prone T2DM.

Differentiating between DKA and HHS, is yet another pitfall in the diagnosis of DKA. The two conditions are hyperglycaemic emergencies, although, with distinct pathophysiologies. HHS, which is rare in children with T1DM and more common in adults with T2DM, is characterized by marked hyperglycaemia and absence of ketosis. Specifically, HHS is characterized by severe hyperglycaemia (glucose > 30 mmol/L or 540 mg/dL), increased serum osmolality (>320 mOsmol/L) due to electrolyte and glucose concentrations, and circulatory volume depletion due to osmotic diuresis, in the absence of ketosis (β-hydroxybutyrate concentrations < 3.0 mmol/L) and acidosis (pH > 7.3 and HCO_3_^−^ > 15 mmol/L) [26]. Insulin concentrations are adequate to inhibit ketogenesis but not sufficient to ensure adequate cellular glucose uptake. Although HHS is less frequent than DKA, it is associated with higher mortality, of up to 20% [19,27]. As with DKA, concurrent illness or physiological stress may precipitate HHS, as a result of an increase in counter-regulatory hormones. The differentiation between the two conditions is necessary as circulatory volume depletion is more severe in HHS compared to DKA; therefore, management of HHS mainly involves more aggressive fluid resuscitation to restore fluid and potassium deficits and reduce hyperosmolality.

In addition, lack of prompt recognition of new-onset T1DM by health-care providers is another pitfall that may increase the risk of DKA [28]. Missed or delayed diagnosis is mainly caused by the presence of clinical symptoms that overlap between T1DM and other, usually more common, medical conditions. Specifically, in children, the clinical symptoms precipitating DKA include: (i) polyuria, i.e., excessive urination, due to osmotic glycosuria with water and electrolyte loss, leading in some cases to enuresis, (ii) polydipsia, i.e., excessive thirst, secondary to polyuria, (iii) polyphagia, i.e., excessive hunger, and (iv) weight loss [29]. Recognizing the hyperglycaemia-induced nature of these symptoms is crucial for a timely diagnosis of a new presentation of diabetes, avoidance of misdiagnosis and prevention of DKA and its associated risks. Frequent misdiagnosis errors include diagnosing a urinary tract infection, attributing increased thirst to heat or increased physical exercise, particularly during the summer, and attributing weight loss to accelerated height gain, particularly during adolescence. Obtaining a thorough medical history can allow distinguishing diabetes-related polyuria from frequent urination caused by a urinary tract infection, which is also characterized by a small urine volume and the urge to urinate. A detailed medical history can also reveal progressively deteriorating, and otherwise unexplained, polydipsia, polyuria, and weight loss. Weight loss and gradually worsening fatigue, which are caused by insulin deficiency and the increase in counter-regulatory hormones that result in lipolysis and muscle lysis in the effort to compensate for intracellular glucopenia and lack of energy, are frequently attributed by parents or children to exercise and increased learning activities.

Once ketosis and acidosis begin to develop, gastrointestinal symptoms are added, including nausea, vomiting and abdominal pain, in more than 60% of patients [7,30]. These symptoms are often misperceived as gastroenteritis, especially in the context of a relatively short history. Therefore, increased index of suspicion is required by the clinician in order to not be misguided. However, the possibility of DKA being triggered by a gastrointestinal tract infection should not be ignored, and this is yet another diagnostic pitfall. Furthermore, in severe metabolic acidosis, abdominal pain may mimic an acute abdomen leading in some cases to the false diagnosis of appendicitis and/or peritonitis. Again, a thorough medical history may reveal pre-existing polyuria, polydipsia, and weight loss; whereas, a careful clinical examination may reveal severe dehydration and circulatory volume depletion presenting as dry mucous membranes, delayed capillary refill time, and tachycardia. Also, measurement of capillary and/or blood glucose concentrations in the presence of such symptoms is of particular importance.

In addition, with the progression of DKA, Kussmaul breathing pattern is observed as a compensatory mechanism for hyperketonaemia and metabolic acidosis, characterized by tachypnoea, deep and laboured breathing. These symptoms may falsely be attributed to a respiratory tract infection or pneumonia. A careful physical examination can differentiate between the two conditions. Notably, DKA is not characterized by symptoms such as cough and fever, or by signs of respiratory distress, with the exception of tachypnoea. In contrast, children with DKA are either normothermic or hypothermic. Also, a fruity odour due to acetone exhalation is typical of DKA. Caution should be raised about the likelihood of DKA being precipitated by a respiratory tract infection.

Finally, if DKA remains undiagnosed, mental status is impaired due to deteriorating dehydration and acidosis, resulting in lethargy or even coma. Excluding CNS infection, such as meningitis or encephalitis, is necessary.

### 4.2. Assessment of Ketosis

Ketone concentrations can be assessed by using the nitroprusside reaction in the urine or serum or by direct measurement of β-hydroxybutyrate in capillary blood [31,32]. Although the nitroprusside test is technically easy, it measures acetoacetate, not β-hydroxybutyrate, which is the main ketone in DKA [33,34]. Acetoacetate accounts for 15–40% of the total ketone concentration; therefore, measuring acetoacetate may underestimate the severity of ketonaemia [11,35]. Also, patients taking anti-epileptics such as valproate may have a false positive nitroprusside urine test [11,36]. Additionally, ketones are detected earlier and cleared faster in blood than urine, which may lead to misinterpretation or overtreatment when controlling ketonuria [23].

### 4.3. Pitfalls Related to the Management of DKA

Fluid and electrolyte replacement and insulin therapy are the cornerstones of DKA management, with the aims of restoring normal circulatory volume, improving glomerular filtration and clearing glucose and ketones from the blood, normalizing glucose concentrations, and correcting acidaemia and electrolyte disturbances [1]. Water and salt deficits need to be replaced and a 10–20 mL/kg bolus with 0.9% normal saline may be required for 1–2 h based on the assessment of hydration status. Careful fluid resuscitation, timely insulin administration, i.e., at least 1 h after the initiation of fluid administration, and appropriate correction of shifting electrolyte imbalances are among the therapeutic challenges. Close monitoring of neurological status and vital signs, including blood pressure, pulse and respiratory rate are essential. Water balance and glucose levels should be documented on an hourly basis and electrolyte concentrations documented every 2–4 h.

### 4.4. Fluid Management

Numerous studies have highlighted the risk of development of cerebral oedema, a potentially devastating consequence of DKA, after initiation of DKA treatment [37,38,39]. The pathophysiologic mechanism underlying diabetic ketoacidosis-related cerebral oedema is controversial. Cerebral oedema was initially thought to occur due to retention of cerebral intracellular osmolytes, resulting in fluid shifts in the intracellular space. According to this theory, during acute hyperglycaemia, osmotically active substances are retained in brain cells to prevent dehydration. With the initiation of treatment, if glucose concentrations decline rapidly, the remaining osmotically active substances create an intracellular osmotic gradient that results in cerebral oedema. Therefore, for years the approach for fluid management involved slow rehydration to mitigate the risks of cerebral oedema [40,41,42]. This dogma is being challenged by newer studies, which offer alternative potential mechanisms of cerebral oedema in DKA, including vasogenic oedema due to blood-barrier destruction, and cytotoxic oedema secondary to ischemia [43]. Recent research has shown that the risk of cerebral injury is neither affected by the infusion rate, nor by sodium chloride concentration [18].

Additionally, restoration of fluid volume is achieved with administration of 0.9% sodium chloride or Ringer’s lactate [22,44]. Typically, plasma glucose concentrations decrease to <11 mmol/L or <200 mg/dL before the resolution of DKA. If insulin infusion is stopped in order to avoid hypoglycaemia before ketonaemia is corrected, ketonaemia and metabolic acidosis will deteriorate. Therefore, when plasma glucose concentrations decrease below 11 mmol/L or 200 mg/dL, dextrose should be added to the replacement fluids to prevent hypoglycaemia but also allow the continuation of insulin administration [7].

### 4.5. Correction of Electrolyte and Acid-Base Disturbances

In the acute phase of DKA, normal serum potassium values are maintained or hyperkalaemia occurs due to metabolic acidosis and the shift of potassium ions from the intracellular to the extracellular space [1]. For each 0.1 unit fall in pH, the serum potassium levels are increased by 0.6 mmol/L [45]. In some cases, in the acute phase of DKA before fluid resuscitation and insulin administration are initiated, serum potassium may exceed 7 mmol/L. However, total body potassium stores are substantially depleted. Insulin deficiency causes potassium to move from the intracellular to the extracellular space. Furthermore, water moves from the intracellular to the extracellular space due to hypertonicity, leading to further loss of intracellular potassium. In addition, the decrease in the circulating volume following osmotic diuresis leads to increased aldosterone concentrations, and subsequently, reabsorption of sodium in the kidney and potassium excretion in the urine, further contributing to potassium loss [46]. Therefore, potassium replacement is almost always required and should be started together with insulin administration, even if serum potassium levels are normal, so that hypokalaemia is prevented. If serum hypokalaemia is present at the time of DKA diagnosis, potassium replacement should be initiated together with resuscitation fluids and the initial insulin infusion should be delayed [35]. In the case of hyperkalaemia, potassium replacement should be held until potassium normalizes, renal function is normal and urinary voiding is intact.

Furthermore, hyponatremia is one of the most common electrolytic disorders in DKA. Hyperglycaemia is associated with water movement from the intracellular to the extracellular space along the osmotic gradient, resulting in a decrease in serum sodium concentrations, also known as pseudohyponatremia [47]. True hyponatremia, particularly in the absence of gradually increasing sodium levels, is associated with poor prognosis and unfavourable clinical outcomes [48]. Therefore, close monitoring of sodium and calculation of corrected sodium concentrations are necessary for the recognition of true hyponatremia, so that higher intravenous concentrations of sodium are administered [48].

Another pitfall in the management of DKA is related to hyperchloraemic metabolic acidosis caused by the administration of large volumes of 0.9% sodium chloride solution during the recovery phase, which includes higher concentration of chloride ions compared to the serum (154 mmol/L vs. 100 mmol/L) [49,50]. Although hyperchloraemic metabolic acidosis is not a dangerous condition, it may delay transition to subcutaneous insulin therapy if the assessment of DKA resolution is based upon serum bicarbonate concentration.

The use of bicarbonate infusion for the management of metabolic acidosis in DKA is also a topic of controversy. Severe acidosis may cause detrimental cardiac and neurologic complications; however, research data have failed to demonstrate therapeutic value of bicarbonate treatment. In addition, bicarbonate use has also been associated with hypokalaemia [51,52]; therefore, bicarbonate therapy is not recommended in children with DKA with the exception of life-threatening hyperkalaemia and severe acidosis (pH < 6.9) with evidence of compromised cardiac contractility [3].

### 4.6. Pitfalls Regarding Insulin Administration

Insulin should be administered at least 1 h after fluid resuscitation has begun. An insulin bolus is not recommended in paediatric patients, as it increases the risk for cerebral oedema [22]. Regular insulin should be administered with a continuous drip at a rate of 0.05–0.1 unit/kg/h and IV Dextrose should be added when serum glucose concentration decreases to 14 mmol/L or 250 mg/dL [20,53]. In the case of blood glucose levels falling below 8 mmol/L or 150 mg/dL, higher concentrations of dextrose may be used, i.e., 10–12.5%. The insulin infusion rate should not be reduced before ketoacidosis is corrected or nearly corrected.

The insulin infusion is discontinued once DKA is resolved. Specifically, the following targets should have been achieved: (i) the patient has no gastrointestinal symptoms and can receive fluids and medications orally, (ii) blood glucose concentrations are less than 11 mmol/L or 200 mg/dL, (iii) serum anion gap is closed or β-hydroxybutyrate is less than or equal to 10.4 mg/dL, (iv) venous pH > 7.3 or serum bicarbonate > 15 mEq/L [46]. Caution is needed during transition from intravenous insulin to subcutaneous injections. The optimal time for the transition is before a meal. Short-acting insulin is administered in the intravenous infusion. Due to its short half-life of 5–7 min, the infusion should be stopped at least 30 min after subcutaneous injection of short-acting insulin or 15 min after injection of rapid-acting insulin [46]. Long-acting/basal insulin should optimally have been started few hours prior to the discontinuation of the insulin infusion. If DKA occurs in the context of pre-existing T1DM, the patient’s basal analogue insulin should be continued alongside the insulin infusion so that rebound hyperglycaemia is prevented after intravenous insulin is stopped.

Of note, in the case of DKA not resolving, the treatment approach should be reassessed. Cannula patency and placement should be checked, administration of intravenous insulin infusion at the correct rate should be confirmed, and the possibility of concomitant pathology, such as infection or sepsis, should also be considered.

### 4.7. Pitfalls Associated with Insulin Pump Use

A significant and progressively increasing proportion of children and adolescents with T1DM are treated with continuous subcutaneous insulin infusion through an insulin pump due to several advantages insulin pumps offer over the multiple dose injection insulin therapy. The more flexible character of insulin administration eliminating the need for several subcutaneous daily injections, the possibility of dose adjustments in small increments and of altered basal rates throughout the day or on different occasions, i.e., during physical activity, menstruation or illness, are some of the main reasons continuous subcutaneous insulin infusion has become the predominant type of treatment for children with T1DM in many countries. In contrast to the basal bolus regimen, which includes basal insulin typically administered once daily and rapid-acting insulin administered during carbohydrate-containing meals, insulin pumps continuously infuse rapid-acting insulin only, but not long-acting background insulin. Due to the possibility of interruption of insulin delivery through the pump, i.e., because of catheter occlusion or device malfunction, concerns have been raised regarding a potentially increased risk of developing DKA [54,55,56,57]. The short half-life of rapid-acting insulin may result in acutely rising blood glucose concentrations and ketone development within 4–6 h. In line with previous findings, a recent 2-year Swedish national survey of children and adolescents with T1DM treated with an insulin pump showed higher rates of mild DKA compared to those treated with insulin injection therapy [58]. Patients on pumps with psychological or social problems and during adolescence are further exposed to secondary DKA. This is an additional “trap” for the unwary and should be highlighted during the education provided by the diabetes team before initiation of insulin pump therapy. Interestingly, other studies have shown opposing results in the incidence rate of DKA when comparing insulin pump use and multiple daily injections. According to large population-based observational studies, the DKA rate has significantly declined in insulin pump users due to technological improvements and appropriate patient training [59,60]. In any case, the use of continuous glucose monitoring (CGM), as well as close monitoring of blood ketones, administration of correction boluses through insulin pens in the case of persisting hyperglycaemia, and frequent reinforcement of DKA prevention education, are some of the necessary and effective measures that should be undertaken [55].

### 4.8. Pitfalls Related to the Use of Sodium-Glucose Transporter 2 (SGLT2) Inhibitors

Although insulin therapy remains the cornerstone of T1DM treatment, adjunctive therapies are gaining popularity over the last few years with the aims of improving glycaemic control and reducing insulin requirements and insulin-associated dose-dependent adverse secondary effects, such as hypoglycaemia and weight gain [61]. Among the studied medications, SGLT2 inhibitors are considered an appealing option due to their glucose lowering effect and favourable side effect profile. It has been shown that SGLT2 inhibitors allow a reduction in total insulin dose by 10–15% in adult patients with T1DM [62]. SGLT2 inhibitors act on the SGLT2 receptors in the proximal tubule of the kidney and inhibit glucose reuptake leading to increased glucose excretion in the urine [62,63]. They also cause natriuresis and their use has been associated with improved long-term kidney outcomes, modest weight loss, and lowering of systolic blood pressure [62,64]. However, restrictions in the use of SGLT2 inhibitors have been widely reported, and adverse outcomes include increased occurrence of genital mycotic infections [65,66] and euglycaemic DKA in patients with T1DM and T2DM [66,67]. Euglycaemic DKA is characterized by anion gap metabolic acidosis, ketonaemia, ketonuria, but normal or modestly elevated glucose concentrations (<250 mg/dL or 13.9 mmol/L), which may delay diagnosis and lead to unfavourable clinical outcomes. Thus far, SGLT inhibitors have not formally been approved for use in children and adolescents with T1DM, however studies evaluating their use are ongoing. Therefore, awareness should be raised regarding the pitfalls associated with their potential use. In 2019, the International Consensus on Risk Management of DKA in Patients with T1DM Treated with SGLT inhibitors published recommendations for the mitigation of DKA risk [68]. The recommendations included starting with the lowest possible dose of SGLT inhibitor, frequent monitoring of ketones after treatment initiation, avoiding more than 10–20% reductions in insulin doses at a time, and educating patients to stop the SGLT inhibitor and manage euglycaemic ketosis by injecting insulin, self-hydrating and eating carbohydrates [68,69]. In addition, selection of patients without risk factors for DKA, such as low carbohydrate diets, recurrent episodes of DKA, and low engagement with their diabetes treatment, is of paramount importance.

### 4.9. Pitfalls Related to DKA Complications

Paediatric DKA is associated with a wide range of complications, with cerebral oedema being the most feared. Cerebral oedema is clinically apparent in 1% of diagnoses of DKA and is associated with a mortality rate of 40–90% [70,71]. It usually develops within the first few hours of initiation of fluid resuscitation, i.e., 7–8 h in approximately 2 out of 3 cases [18]; whereas, in the remaining cases it occurs up to 28–30 h after fluid resuscitation and initiation of insulin treatment [43]. It has been reported, however, that cerebral oedema may rarely occur prior to or up to 60 h after treatment initiation, which highlights the need for vigilance and continuous monitoring of the patients’ mental status [43]. Risk factors for cerebral oedema include severe acidosis, severe dehydration, elevated blood pressure and markedly elevated BUN [3]. As already mentioned, rapid IV fluid resuscitation is discouraged, however no difference was found in the neurological outcomes between different rates of IV fluid administration in a recent study [18]. Warning clinical symptoms and signs include altered mental status, such as lethargy, irritability and confusion, onset of headache, progressively worsening vomiting or vomiting after beginning of treatment, urinary incontinence, specific neurological signs, i.e., cranial nerve palsies, and Cushing Triad (bradycardia, irregular respirations, hypertension). New headache, recurrence of vomiting should raise suspicion, particularly in the presence of severe ketoacidosis and hypertension [21].

Clinical identification of cerebral oedema is confounded due to similar clinical presentations caused by other medical conditions; alterations in mental status could be attributed to severe dehydration and acidosis, vomiting could be attributed to acidosis and ketosis, and urinary incontinence to polyuria [72,73].

Another pitfall is that cerebral oedema may not initially be visible on CT scan of the brain; therefore, if suspicion is high, treatment should be started [74].

### 4.10. Acute Kidney Injury (AKI)

Among the most common complications of DKA in children and adolescents is AKI, which occurs in 43% to 64% of DKA episodes in children [75,76]. In one-fourth of DKA episodes, AKI is severe, suggesting severe volume depletion [77] and highlighting the need for a delicate balance between treating severe hypovolaemia but also avoiding excessive fluid replacement that may increase the risk for cerebral injury [78]. Awareness about this complication and early recognition of AKI are important also because potassium repletion should not be started if renal function is impaired and should be withheld until urine output is documented [46].

Moreover, AKI has been associated with a substantially increased hazard rate for development of microalbuminuria and contributes to the development of diabetic kidney disease [79]. Therefore, timely fluid resuscitation is important for the prevention of AKI and for ameliorating the associated short- and long-term consequences.

Additional complications of DKA include hypokalaemia, hypoglycaemia, venous thrombosis, pancreatic enzyme elevations, rhabdomyolysis, pulmonary oedema, and cardiac arrhythmias. Prevention of all DKA-related complications, involves primarily prevention of DKA itself.

A summary of the diagnostic and therapeutic pitfalls during DKA management is presented in Table 2 and Table 3.

In conclusion, this review is the first to summarize the diagnostic and therapeutic pitfalls which may cause delays in the identification and management of DKA and its related complications. The present study delineates the importance of awareness and a high suspicion index so that this diagnosis is considered in the presence of potentially related symptoms. Accurate and rapid diagnosis, prompt intervention, and meticulous monitoring are of major importance to break the vicious cycle of life-threatening events and prevent severe complications during this potentially fatal medical emergency.

## Figures and Tables

**Figure 1 diagnostics-13-02602-f001:**
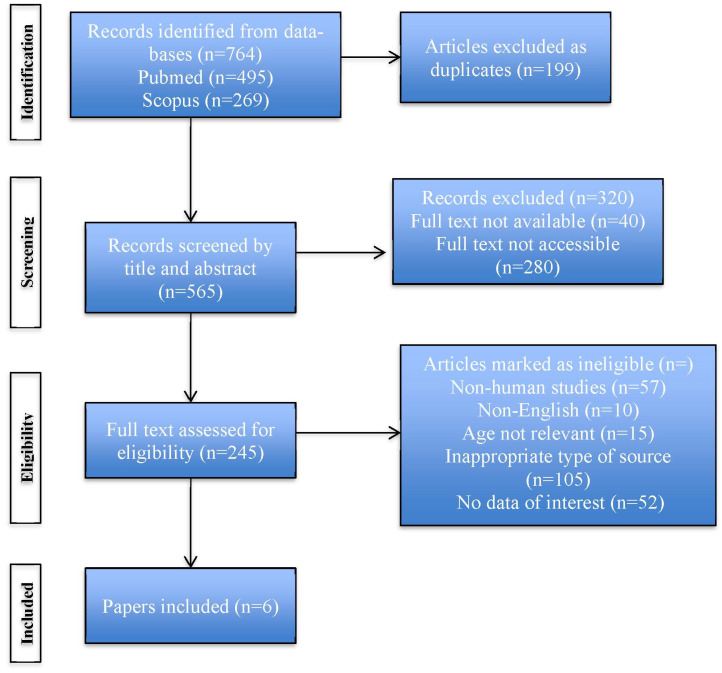
PRISMA flow diagram describing the stages of the search strategy.

**Table 1 diagnostics-13-02602-t001:** Summary of included articles.

Author(s)	Title	Year	Type	Country
Kuppermann, N., et al. [18]	Clinical trial of fluid infusion rates for pediatric diabetic ketoacidosis	2018	Clinical trial	USA
Pasquel, F.J., et al. [19]	Hyperosmolar hyperglycemic state: a historic review of the clinical presentation, diagnosis, and treatment	2014	Review	USA
Rameshkumar, R., et al. [20]	Low-dose (0.05 Unit/kg/h) vs. Standard-Dose (0.1 Unit/kg/h) Insulin in the Management of Pediatric Diabetic Ketoacidosis: A Randomized Double-Blind Controlled Trial	2021	Randomized double-blind controlled trial	India
Dunger, D.B., et al. [21]	ESPE/LWPES concensus statement on diabetic ketoacidosis in children and adolescents	2004	Review	UK
Hsia, D.S., et al. [22]	Fluid management in pediatric patients with DKA and rates of suspected clinical cerebral edema	2015	Comparative study	USA
Vanelli, M., et al. [23]	Clinical utility of beta-hydroxybutyrate measurement in the management of physiological ketosis at home in children under 5	2019	Comparative study	Italy

**Table 2 diagnostics-13-02602-t002:** Pitfalls in the diagnosis of DKA.

Pitfalls	Parameters to Be Considered	Recommendations
**Diagnosis of DKA**		
Diagnosis of correct type of diabetes	Differentiation between T1DM and T2DM	Assess for insulin concentrations and autoimmunity [8]
Diagnosis of correct pathophysiological mechanism	Differentiation between T1DM and HHS	Assess for marked hyperglycaemia in the absence of ketosis [26]
Avoid misdiagnosis errors	Exclusion of urinary tract infection, pneumonia, asthma exacerbation, gastroenteritis, acute abdomen, CNS infection, increased thirst due to heat or increased physical exercise, weight loss due to accelerated height gain	Accurate medical history, physical examination, capillary or blood glucose measurement, blood tests [7,29,30]
Co-existence of precipitating factors	Exclusion of infection, sepsis	Accurate medical history, blood tests [7,8]
Assessment of ketosis	The severity of ketonaemia is underestimated by urine acetoacetate. Blood ketones are detected earlier and cleared faster than urine ketones.	Assessment of blood ketones preferable to assessment of urine ketones [11,33,34,35].

**Table 3 diagnostics-13-02602-t003:** Pitfalls in the management of DKA.

Pitfalls	Parameters to Be Considered	Recommendations
**Treatment of DKA**		
Fluid management/infusion rate	Equilibrium between restoration of normal circulatory volume and the risk of cerebral oedema	No strict fluid resuscitation, as cerebral injury is not affected by the infusion rate [23,37,38,39,40,41]
Fluid management/type of solutions	Equilibrium between prevention of hypoglycaemia and deterioration of ketonaemia and acidosis	Dextrose should be added in the replacement fluids to prevent hypoglycaemia and allow continuation of insulin infusion [7]
Electrolyte correction	Total body potassium stores depleted regardless of serum potassium values	Potassium replacement almost always required alongside insulin infusion initiation [22,35]
	Identification of true hyponatremia versus pseudohyponatremia	Close monitoring of sodium and calculation of corrected sodium concentrations [46]
Correction of acid base disturbances	Hyperchloraemic acidosis due to large volumes of 0.9% sodium chloride solution	Assessment of DKA resolution not only by serum bicarbonate concentration [47,48]
	Bicarbonate use for the management of metabolic acidosis not supported by research data	Bicarbonate use only in life-threatening hyperkalaemia or severe acidosis (pH < 6.9) [3]
Insulin administration	Timely intravenous insulin administration	At least 1 h after initiation of fluid resuscitation; Insulin bolus not recommended [22]
	Correct time of insulin infusion discontinuation	Insulin infusion to be discontinued only after DKA is resolved [22]
	Transition from intravenous to subcutaneous insulin	Transition optimally before a meal; Subcutaneous insulin injection prior to discontinuation of insulin infusion [22]
Insulin pump users	Risk of pump failure	Use of CGM, monitoring of blood ketones, administration of correction boluses through insulin pens in the case of persisting hyperglycaemia [58]
DKA complications	Risk of cerebral oedema	Awareness regarding warning clinical symptoms and signs [69,70]
	Risk of acute kidney injury (AKI)	Timely management of hypovolaemia; Careful potassium replacement if AKI occurs [22]

## Data Availability

No new data were created or analyzed in this study. Data sharing is not applicable to this article.

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
