# Peer review of "Diabetic Ketoacidosis in Children and Adolescents; Diagnostic and Therapeutic Pitfalls"

_diagnostics, 2023, doi:10.3390/diagnostics13152602_

Round 1

Reviewer 1 Report

1.                   The paragraph regarding insulin pump use and DKA should be more extensive, having in mind increasing usage of insulin pumps in paediatric T1D

2.                   The authors should emphasise briefly the role of SGLT2i as potential cause of DKA, having in mind that it is approved for children

3.                   Please, split Table 1 into 2 Tables

Author Response

We thank the Editor and the reviewers for the constructive comments, which we believe we have addressed and which have helped us improve our manuscript. The revisions are highlighted in yellow.

Point-to-point response to reviewers’ comments

Reviewer 1

Comments and Suggestions for Authors

  1. The paragraph regarding insulin pump use and DKA should be more extensive, having in mind increasing usage of insulin pumps in paediatric T1D.

Response: We thank the reviewer for the comment. We have extended the paragraph regarding insulin pump, as per the reviewer’s suggestion.

  1. The authors should emphasise briefly the role of SGLT2i as potential cause of DKA, having in mind that it is approved for children.

Response: We thank for the comment. We have made a brief reference to the role of SGLT2i as a potential cause of DKA in the revised version of the manuscript.

  1. Please, split Table 1 into 2 Tables.

Response: We have split Table 1 into two tables, Tables 2 and 3 in the revised version, as suggested.

Reviewer 2 Report

Comments to the Authors,

Minor comments:

·       It is preferable to use the abbreviation sin details in the first time use e.g. Diabetic ketoacidosis (DKA), type 2 diabetes (T1DM). 

·       It is preferable to unify the reference style with unifying the number of authors listed in details in all the references.

·       In the table of recommendations, it would be nice to add the reference of each recommendation.

Major comments:

·       The manuscript requires English revision because of some spelling and linguistic errors.

Methodology:

·       The study needs to describe the methodology of data search and extraction and selection of high quality sources.

·       The study lacks results and discussion sections. 

English editing required

Author Response

We thank the Editor and the reviewers for the constructive comments, which we believe we have addressed and which have helped us improve our manuscript. The revisions are highlighted in yellow

Point-to-point response to reviewer's comment

Reviewer 2

Comments to the Authors,

Minor comments:

  • It is preferable to use the abbreviations in details in the first time use e.g. Diabetic ketoacidosis (DKA), type 2 diabetes (T1DM).  

Response: We thank the reviewer for the comment. The abbreviations have now been used in detail in the first time used.

  • It is preferable to unify the reference style with unifying the number of authors listed in details in all the references.

Response: We thank for the comment. In the revised version of the manuscript the reference style is unifying the number of authors.

  • In the table of recommendations, it would be nice to add the reference of each recommendation.

Response: The reference(s) of each recommendation have been added to the current version of the manuscript. We thank for the comment.

Major comments:

  • The manuscript requires English revision because of some spelling and linguistic errors.

Response: We thank the reviewer for the comment. The manuscript has been edited by a native English speaker.

Methodology: 

  • The study needs to describe the methodology of data search and extraction and selection of high quality sources. 

Response: The methodology of data search has been added and the PRISMA guidelines have been used.

  •  The study lacks results and discussion sections. 

Response: Results and Discussion sections have been added to the text.